# Expression characteristics and regulatory mechanism of *Apela* gene in liver of chicken (*Gallus gallus*)

**Wenbo Tan**[1☯], **Hang Zheng**[2☯], **Dandan Wang**[2], **Fangyuan Tian**[2], **Hong Li**[2], **Xiaojun Liu**[2,3]*

1 College of Medicine, Shihezi University, Shihezi, Xinjiang, China, 2 College of Animal Science and Veterinary Medicine, Henan Agricultural University, Zhengzhou, China, 3 College of Animal Science and Technology, Shihezi University, Shihezi, Xinjiang, China

☯ These authors contributed equally to this work.
* xjliu2008@hotmail.com, 603679737@qq.com

**Data Availability Statement:** All relevant data are within the paper and its Supporting Information files.

## Abstract

Apela, a novel endogenous peptide ligand for the G-protein-coupled apelin receptor, was first discovered and identified in human embryonic stem cells in 2013. Apela has showed some biological functions in promoting angiogenesis and inducing vasodilatation of mammals by binding apelin receptor, but little is known about its expression characteristics and regulatory mechanism in chicken. In the present study, the coding sequences of *Apela* in chicken was cloned. The evolution history and potential function of *Apela* were analyzed. Subsequently, the spatiotemporal expression characteristics of chicken *Apela* were investigated. Furthermore, the regulatory mechanism of *Apela* mRNA responding to estrogen was explored by *in vitro* and *in vivo* experiments. The results showed that the length of the CDs of *Apela* mRNA was 165 bp and encoded a protein consisting of 54 amino acids residues with a transmembrane domain in chicken. The *Apela* was derived from the same ancestor of *Apelin*, and abundantly expressed in liver, kidney and pancreas tissues. The expression levels of *Apela* in the liver of hens were significantly higher at the peak-laying stage than that at the pre-laying stage ($p \leq 0.05$). The *Apela* mRNA levels were significantly up-regulated in primary hepatocytes treated with 17β-estradiol ($p \leq 0.05$), and could be effectively inhibited by estrogen receptor antagonists MPP, ICI 182780 and tamoxifen. It indicated that chicken *Apela* expression was regulated by estrogen via estrogen receptor α (ERα). In individual levels, both the contents of TG, TC and VLDL-c in serum, and the expression of *ApoVLDLII* and *Apela* in liver markedly up-regulated by 17β-estradiol induction at 1mg/kg and 2mg/kg concentrations ($p \leq 0.05$). This study lays a foundation for further research on Apela involving in hepatic lipid metabolism.

## Introduction

The endogenous peptide apelin is identified as a novel adipokine to regulate fat metabolism in mammals, exerting its biological function by binding the G-protein-coupled apelin receptor

**Funding:** This research was supported by the National Natural Science Foundation of China-NSFC-Henan joint grant (grant number U1704233).

**Competing interests:** The authors declare no conflicts of interest.

(APJ) to activate downstream pathways [1]. The apelin-APJ system involves in many physiological and pathological processes, such as energy metabolism [2], diabetes [3], obesity [1], and cardiovascular development [4]. Since the apelin was first isolated and identified for the human APJ receptor [5], it was thought of as the only endogenous ligand of the APJ receptor. But in fact, the knockout of mice apelin don't cause early embryonic developmental defects [6], which contradicts the inactivation of APJ, the loss of which seriously influences the embryonic lethality [7]. Subsequently, two independent groups of researchers discovered and characterized a gene called Apela [8, 9], which has little sequence similarity to Apelin, but can bind to the apelin receptor APJ to exert unique biological functions as the second endogenous ligand [10]. Thereby, it provides the answer why the inactivation of APJ instead of the lack of apelin causes defects in cardiac development and increased birth mortality in knockout mice [7].

As the two ligands of APJ receptor, the structure and the form of biological activity of endogenous/synthetic apelin peptides have been already relatively clear [5, 11–13]. And the functional study of apelin-APJ, linking with Lipid synthesis and metabolism, have been widely reported, especially in mammals. For instance, apelin binding APJ, could inhibit adipogenesis of pre-adipocytes through MAPK kinase/ERK dependent pathways and could also suppress basal lipolysis in mature adipocytes through AMP kinase dependent enhancement of perilipin expression [14]. Apelin-APJ system activates the C-jun N-terminal kinase (JNK) protein, thereby promotes the fat-catabolism that is dependent on fatty acid synthase (FAS) in liver cells [15]. Apelin, as a relatively new adipokine, co-expresses with APJ receptor in adipocytes. And the injection of apelin causes a decrease of triglycerides (TG) in serum level of mice [16]. Compared to the study of *apelin*, the fundamental and functional research of *Apela* are relatively lagged and rare as the second endogenous ligand of APJ.

Based on recent researches, *Apela* consists of three exons and contains a conserved open reading frame (ORF) that encodes a predicted polypeptide of 54 amino acids (aa) with a signal peptide of 22aa [8]. And the mature Apela peptides are predicted to include three forms with 32, 21, and 11 amino acids based on peptidase cleavage [8, 9]. Phylogenetic analysis revealed Apela possesses a pair of conserved cysteines, and the last 13 residues are nearly invariant in all vertebrate species [8], which suggests a functional conservation of Apela-APJ pathway. When it comes to functionality, although *Apela* mRNA is expressed in human stem cells, prostate, and kidney [8, 17], and Apela can promote angiogenesis and induce vasodilatation in mouse aorta through activating signal transduction pathways [17], and use as a pre-warning biomarker and a novel therapeutic target against progression of heart failure [18]. In addition, Apela can function as a regulatory RNA to regulate p53-mediated DNA damage-induced apoptosis (DIA) of mouse embryonic stem cells (mESCs) [19].

In chicken, liver plays a critical role in lipid synthesis, degradation, and transport [20]. Especially during the transition from the sexually immature to laying phases, the liver of laying hens undergoes many metabolic changes to support vitellogenesis [20, 21]. In the process, many lipid-metabolic genes, including those involved in lipogenesis, holding, breaking down and transporter functional genes, are sharply up-regulated or down-regulated to help achieve these metabolic changes [22]. Meanwhile, estrogen, as an important inducer of certain liver genes, rises significantly in the sexually mature hens compared to the sexually immature hens [23]. And most of those estrogen-induced genes involve in hepatic fatty acid synthesis and the assembly of triglyceride-rich lipoprotein particles to lead to hypersecretion of these lipoproteins into the circulation [24, 25].

Our previous RNA-seq study showed that *Apela* mRNA was significantly overexpressed in the liver of laying hens comparing with the juvenile hens [26], which implies that *Apela* may play an important role in hepatic lipid metabolism in laying hens. Though chicken *Apela* has been fully cloned and functionally characterized [27], its expression characteristics and

regulatory mechanism remains unclear. In this study, evolutionary history and dynamic expression patterns of chicken *Apela* were investigated by phylogenetic tree and qRT-PCR analyses, respectively. Furthermore, the expression regulation of *Apela* was investigated by in vivo and vitro experiments. This study will lay a foundation for further research on *Apela* involving in hepatic lipid metabolism.

## Materials and methods

### Animals and sample preparation

The birds used in the study are HY-LINE hens. They were raised under the same condition with normal feeding procedure. The liver tissues of hens at the age of 1 day, 1, 10, 15, 20, 30 and 35 weeks (6 birds for each point) were collected, respectively. Meanwhile, ten other tissues including ovary, lung, heart, spleen, kidney, duodenum, glandular stomach, pancreas and pectorales of hens were also collected at the age of 10 and 30 weeks, and stored at -80˚C until use.

Forty-eight 10-week-old HY-LINE hens with similar body weights were randomly divided into four groups (12 birds for each group), including three estrogen-treated groups at different concentrations (0.5, 1.0, and 2.0 mg/kg body weight of 17β-estradiol) and one control group (olive oil only). 17β-estradiol (Sigma-Aldrich, St. Louis, MO, USA) was dissolved in olive oil. The mode of administration was intramuscular injections. After injection for 12 h and 24h, the blood samples of each group was collected from wing veins (6 birds at each observing time) and was left at room temperature for about 1.5 hours to separate the serum. Then, the birds were euthanized, and the liver tissues were isolated. The serum and tissue samples were stored as described above.

The animal use and care protocols for these experiments were approved by the Institutional Animal Care and Use Committee (IACUC) of Henan Agricultural University Zhengzhou, P.R. China (Permit Number: 11–0085).

### Blood plasma biochemical index detection

The blood plasma biochemical index of associated with lipid synthesis and metabolism mainly include triglyceride (TG), low-density lipoprotein cholesterol (LDL-c), high-density lipoprotein cholesterol (HDL-c), total cholesterol (TC) and very low-density lipoprotein cholesterol (VLDL-c) [25, 28]. Among them, the cotent of first four index in serum samples were detected using a blood analyzer (Hitachi 7100), while the cotent of VLDL-c was indirectly calculated by the Friedewald formula: VLDL-c = TC-HDL-c-LDL-c [26, 27, 29, 30].

### Isolation and culture of chicken primary hepatocytes

Fertilized eggs without specific pathogen for the experiment were purchased from Beijing Meiliyaweitong Experimental Animal Technology Co. Ltd. (Beijing, China). Fertilized eggs were incubated to 18 embryo age in thermostatic wet incubator. The embryonic hepatocytes were isolated according to the method as previously described [28, 31]. The seed plate and culture method of hepatocytes were following by our previous studies [29, 32].

### Stimulation of chicken primary hepatocytes

To further confirm estrogen regulating *Apela* expression in vitro, 17β-estradiol, dissolved in ethyl alcohol, were used to stimulate primary hepatocytes. The hepatocytes were starved for 6 h when the confluence reached 80%, then treated with different concentrations of 17β-estradiol (0, 25, 50, 100 nM) for 12 h (six repeats for each treatment). The control groups (0 nM 17β-estradiol) were treated with solvent ethyl alcohol only.

To determine which estrogen receptor subtypes response to estrogen regulating *Apela* expression, 17β-estradiol and estrogen receptor antagonists (ICI 182780, tamoxifen and MPP) were used for stimulating primary hepatocytes. ICI 182780 and tamoxifen are antagonists of ERα and ERβ, tamoxifen also induces GPR30, and MPP (methyl-piperidino-pyrazole) is a highly selective antagonist of ERα. Before treatment, estrogen receptor antagonists were dissolved in DMSO. Similarly, after 6 h of starvation, the primary hepatocytes were co-treated with 17β-estradiol and different estrogen receptor antagonists. In details, the experimental treatment was divided into 5 groups with 6 biological replications of each treatment. The treatment group of estrogen receptor antagonists (group 1, 2, 3) were firstly treated with a final concentration of 1 μM MPP, 1 μM tamoxifen, 1 μM ICI 182,780 for 4 h, respectively, and followed by a treatment of 100 nM 17β-estradiol for 12 h. Estrogen treatment group (group 4) was firstly treated with 1 μM solvent DMSO for 4 h and then a same dose of 17β-estradiol for 12 h. The control group (group 5) was treated with the two corresponding solvents mentioned above. The cells were collected and stored at -80˚C for further use.

## RNA extraction and cDNA synthesis

RNAiso Plus reagent (Takara, Dalian, China) were used to extract the total RNA of tissues and cells. The RNA integrity and quality were tested through gel electrophoresis. The total RNA concentrations were detected by a Nanodrop 2000 spectrophotometer (Thermo Fisher Scientific, Waltham, MA, USA). cDNA was synthesized using a PrimeScript™ RT Reagent kit with gDNA Eraser (Takara) according to the manufacturer's protocols, and stored at -20˚C.

## Cloning of chicken Apela gene

To clone the coding sequence (CDs) of *Apela*, the PCR primers (Table 1) were designed with Primer 5.0 software, according to the predicted Gallus gallus *Apela* sequence (GenBank: XM_015285307). PCR reaction volume was 20 μL, including 10 μL of 2× pfu MasterMix (TiangenBiotech Co. Ltd., Beijing, China), 0.8 μL forward / reverse primers (10 μM), 6.4 μL of deionized water and 2 μL of first-strand cDNA. The PCR procedure was set as follows: 95˚C for 4 min; 30 cycles at 95˚C for 30 s, 60˚C for 30 s, and 72˚C for 30 s; followed by 72˚C for 10 min. The PCR products with correct target fragments were sequenced Sangon Biotech, Co., Ltd. (Shanghai, China) with three times independently.

## Phylogenetic analysis

The amino acid sequence alignments of *Apela* and *Apelin* were performed using Clustal W. And the phylogenetic tree was constructed by using the neighbor-joining method in

**Table 1. Primer sequences used in this study.**

| Primer name | Primer sequence (5'-3') | Product length (bp) | Purpose |
|---|---|---|---|
| *Apela*-F1 | CTTCTGTACACACGCGGACC | 480 | Fragment PCR for obtaining CDs |
| *Apela*-R1 | CCTTCCCCGTTTCTCCCTTC | | |
| *ApoVLDLII*-F | CAATGAAACGGCTAGACTCA | 108 | qRT-PCR |
| *ApoVLDLII*-R | AACACCGACTTTTCTTCCAA | | |
| *Apela*-F2 | TGGTGTCTGACTTCAAGGACT | 188 | qRT-PCR |
| *Apela*-R2 | CAGGTTGGCCGGTCTCTG | | |
| *β-actin*-F | GAGAGAAGATGACACAGATC | 116 | qRT-PCR |
| *β-actin*-R | GTCCATCACAATACCAGTGG | | |

MEGA7.0 [30, 33]. The structure prediction for amino acid sequence of chicken apela was implemented using the SMART software (http://smart.embl-heidelberg.de/).

## qRT-PCR

qRT-PCR primers for *Apela*, *ApoVLDLII*, and *β-actin* were designed according to their corresponding sequences (GenBank: XM_015285307.1, NM_205483.2 and NM_205518.1, respectively) (Table 1). *ApoVLDLII* (very low density apolipoprotein II), which was an estrogen-dependent activation and expression gene in avian [31, 32, 34, 35], was used as a positive control. *β-actin* was used as the endogenous control gene. The reaction volume referred to Zheng et al. [29, 32]. The qRT-PCR reaction was implemented on LightCycler®96 real-time PCR system (Roche Applied Science, Penzberg, Germany), according to the procedure (preincubation at 95˚C for 300s; followed by 35 cycles of 95˚C for 30 s, 60˚C for 30 s, and 72˚C for 30 s).

## Statistical analysis

The mRNA level of *Apela* and *ApoVLDLII* relative to *β-actin* was calculated by the $2^{-\Delta\Delta CT}$ method. The variance analysis among groups were carried out using One-way ANOVA on SPSS version 20.0. $p \leq 0.05$ was considered statistically significant. All values are presented as mean ± SEM.

## Results

### Sequence analysis of chicken Apela gene

The cDNA synthesized from the RNA of liver tissue was used as the template to clone the *Apela* gene (Fig 1A). The sequencing results of PCR product showed that the length of the CDs of *Apela* mRNA was 165 bp, which encoded a protein consisting of 54 amino acide residues (Fig 1B), identical to the nucleotide sequence and amino acid sequence of chicken Apela (KX017223 and APU52335.1, respectively) reported before. The functional domain analysis showed that Apela was composed of two domains including the low-complexity region (LCR) domain and the transmembrane region domain (Fig 1C). The transmembrane region domain was located at 7–26 aa. The LCR was located at 2–18 aa. The amino acid sequences alignment of Apela among chickens, humans and rats indicated that the sequences of Apela shared 59% and 57% identify to human and rat, respectively (Fig 1D).

### Phylogenetic analysis of Apela and Apelin

Phylogenetic analysis was carried out based on the Apela and Apelin amino acid sequences alignment among different species, including mammals (Homo sapiens, Pan troglodytes, Microcebus murinus, Bos taurus, Capra hircus, and Mus musculus), avians (Gallus gallus, Coturnix japonica, Parus major, and Taeniopygia guttata), reptiles (Gekko japonicus), amphibians (Xenopus tropicalis and Gavialis gangeticus) and fish (Danio rerio). The GenBank ID of Apela and Apelin in these species were listed in Table 2. The phylogenetic tree revealed that both Apelin and Apela genes from different species were classified into the same cluster except Apelin from Pan troglodytes was classified into Apela cluster, and *Apelin* and *Apela* are paralogous genes which may originate from the same ancestor (Fig 2).

### Spatiotemporal expression of chicken Apela gene

To understand the expression distribution of *Apela*, *Apelin* and *Apelin receptor* in chicken, the relative mRNA expression levels of the gene in different tissues of 10-week-old and 30-week-old chickens were analyzed by qRT-PCR. The results showed that *Apela* gene was extensively

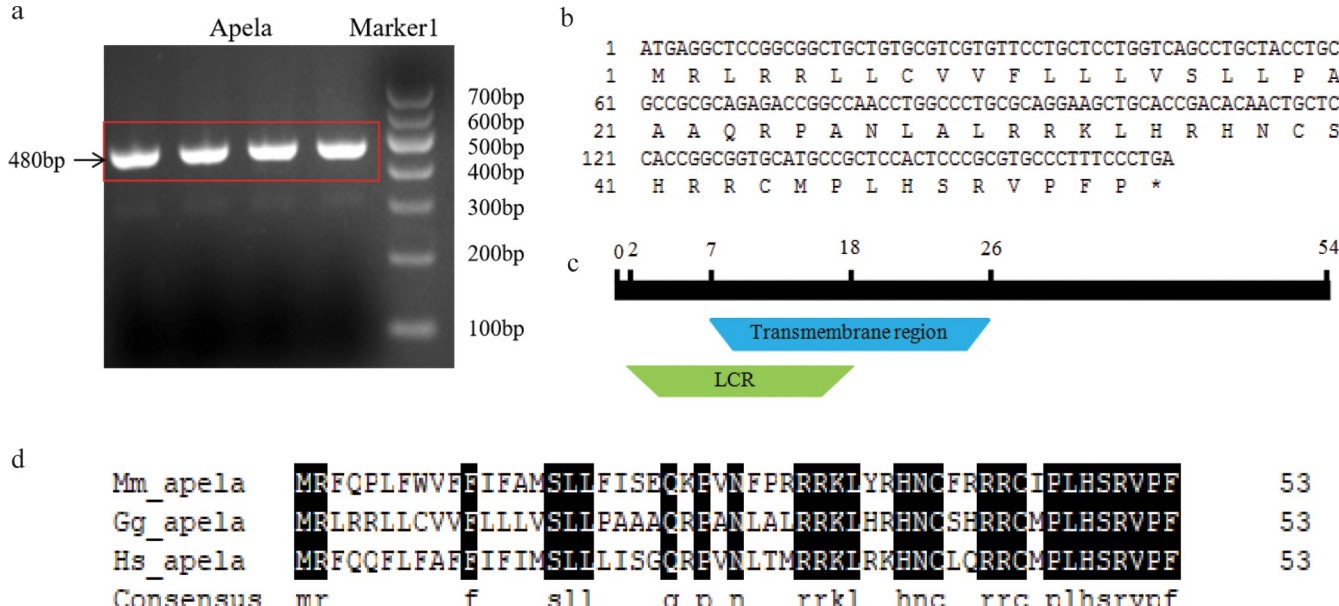

**Fig 1. The cloning and protein structure domain analysis of chicken *Apela*.** (a) The cloning of chicken *Apela*. The electrophoretic band of *Apela* was in the red box. (b) The coding sequence of chicken *Apela* and its corresponding amino acid sequences. The relative position of the first base in each row encoding the nucleotide sequence of CDs, and the relative position of first corresponding amino acid in each row is also given. The asterisk indicates the termination codon. (c) Apela amino acid sequences structure domain analysis. The black stripe represents Apela amino acid sequence, green color means Low-Complexity Region (LCR), blue means transmembrane region. The numbers show the relative positions of the amino acid (aa) in Apela amino acid sequence. The transmembrane region domain was located at 7–26 aa and the LCR was located at 2–18 aa. (d) The amino acid sequences alignment of Apela among chickens, humans and rats. Dark shadows indicate that amino acids are the same in all three species.

expressed in all tissues tested, and the expression levels were relatively higher in liver, pancrea and kidney compared to the other tissues at the two stages (Fig 3). Interestingly, *Apela* was sharply up-regulated only in liver of 30-week-old hens, compared with 10-week-old hens.

**Table 2. The GenBank ID of Apela and Apelin in different species.**

| Species | Apela GenBank ID | Apelin GenBank ID |
|---|---|---|
| Gallus gallus | NP_001295179.1 | XP_015133872.1 |
| Bos taurus | NP_001295178.1 | NP_776928.1 |
| Homo sapiens | NP_001284479.1 | AAH21104.2 |
| Microcebus murinus | XP_012591029.1 | - |
| Mus musculus | NP_001284483.1 | AAH20015.1 |
| Gavialis gangeticus | XP_019377287.1 | - |
| Parus major | XP_015480846.1 | - |
| Capra hircus | XP_013832670.1 | - |
| Xenopus tropicalis | XP_012811093.1 | - |
| Pan troglodytes | XP_009446783.1 | JAA33927.1 |
| Coturnix japonica | XP_015716495.1 | XP_015715548.1 |
| Gekko japonicus | XP_015274400.1 | - |
| Taeniopygia guttata | XP_012428876.1 | - |
| Danio rerio | - | AAY46798.2 |
| Meleagris gallopavo | - | XP_010713361.1 |
| Alligator Mississippians | - | KYO45149.1 |
| Falco peregrinus | - | XP_005236836.1 |

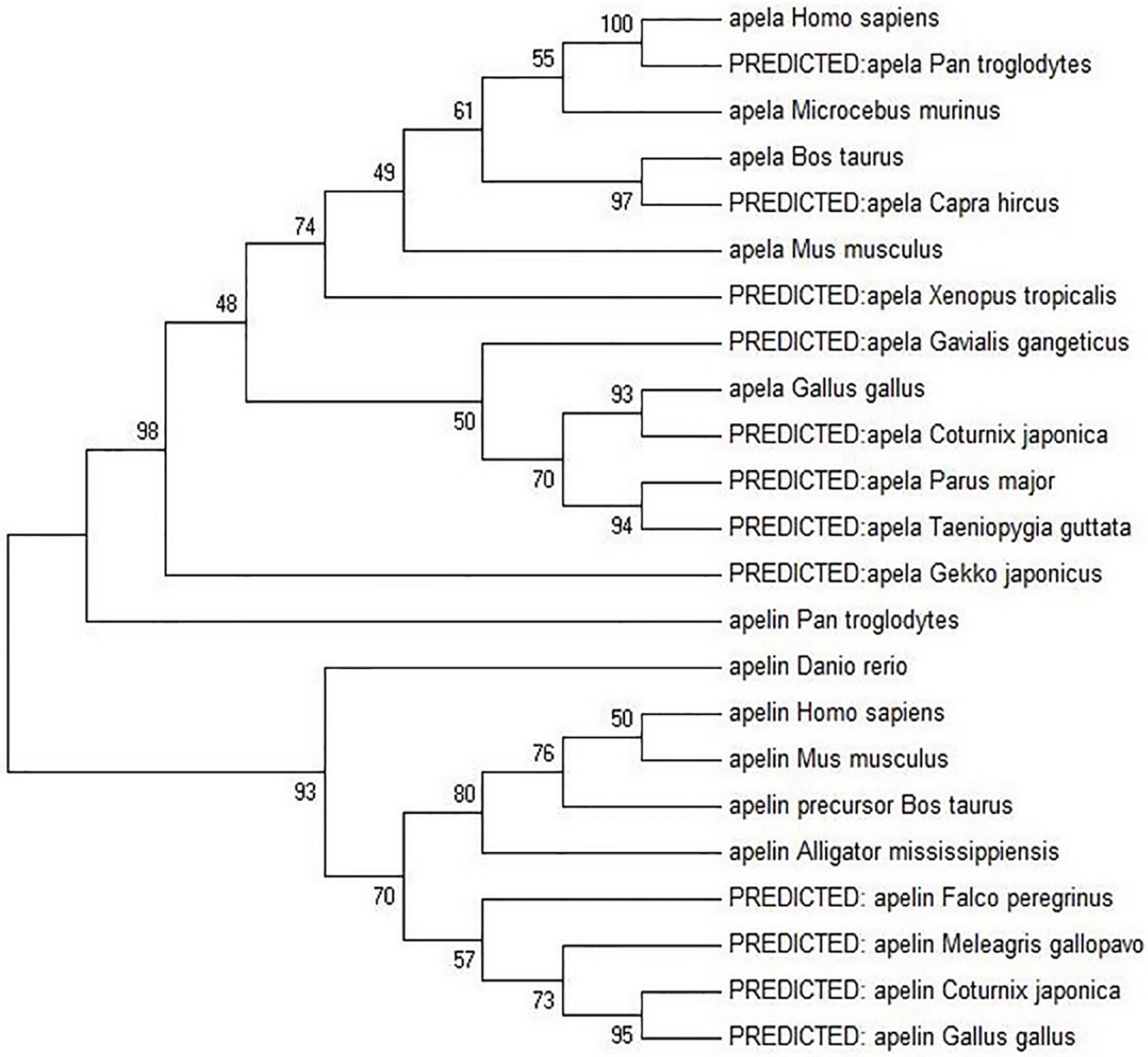

**Fig 2. Phylogenetic relationships of Apela and Apelin among different species.** The amino acid sequence alignments were conducted using ClustalW. The evaluation of statistical confidence in the nodes was based on 1000 bootstrap replicates. Branches with < 50 % bootstrap values were collapsed. Branch lengths reflect evolutionary divergence. The GenBank ID of amino acid sequences used in the evolutionary analyses of vertebrate Apela and Apelin were listed in Table 2.

Furthermore, the relative expression of *Apela* gene in hen livers at different development phases were investigated. It was revealed that, compared to the sexually immaturity (1d, and 1w-20w), the *Apela* mRNA was significantly up-regulated at peak-laying period (30w and 35w) ($p \leq 0.05$), while there is no significant change among 1d, 1w, 10w,15w and 20w ($p > 0.05$), and between 30w and 35w ($p > 0.05$) (Fig 4). However, the expression of *Apelin* and *Apelin receptor* were significantly decreased at the peak-laying stage in comparison with that in pre-laying stage (S1 Fig).

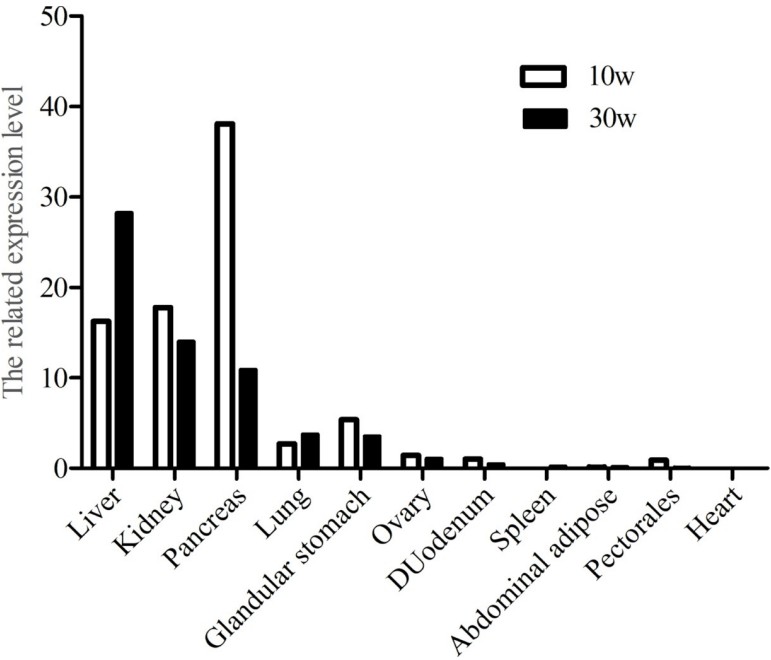

**Fig 3. mRNA expression levels of *Apela* gene relative to *β-actin* in 11 tissues of 10-week-old juvenile female chicken (□) and 30-week-old sexually mature laying female chicken (□).** The tissues were as follows: liver, kidney, pancreas, lung, glandular stomach, ovary, duodenum, spleen, abdominal adipose, pectorales and heart.

## Expression regulation of Apela gene by estrogen *in vitro*

Consider that there are large increase in lipid synthesis of liver and estrogen levels in laying period, the experiment explore the effect of 17β-estradiol on hepatic *Apela* expression *in vitro*. the *Apela* mRNA levels were investigated in chicken primary hepatocytes treated with different concentrations of 17β-estradiol for 12 h. The results showed that *ApoVLDLII*, a positive control gene of estrogen response, was significantly up-regulated expression in a dose-dependent manner (p ≤ 0.05) (Fig 5A), indicating that exogenous 17β-estradiol exerted the biological function of estrogen in the cell level. *Apela* in chicken primary hepatocytes also exhibited a significantly up-regulated expression when 17β-estradiol concentrations achieved at 50 nM and 100 nM (p ≤ 0.05) (Fig 5B). Combining the expression changes of *ApoVLDLII* and *Apela*, 100nM 17β-estradiol concentrations was the optimal concentration that induced *Apela* expression. In contrary, the expression of *Apelin* and *Apelin receptor* exhibited down-regulated trend after 17β-estradiol treated primary hepatocytes (S2 Fig).

To further identify which estrogen receptor subtype responded to estrogen regulation of *Apela* expression, firstly, we checked the expression levels of ER α and ER β in the liver of pre-laying hens (20 weeks old) and peak-laying hens (30 weeks old). The results suggested that the expression levels of ER α and ER β increased with sexual maturation (S3A and S3B Fig). Then, we treated the primary embryonic hepatocytes of chicken with 17β-estradiol or co-treated with 17β-estradiol and different ER antagonists (Fig 6A and 6B). The results showed that compared to the control group, the mRNA levels of *ApoVLDLII* (the positive control gene) increased significantly in 100 nM 17β-estradiol treatment group (p ≤ 0.05), and then decreased significantly to the level of control group in the co-treatment group of 17β-estradiol and ER antagonist MPP, tamoxifen or ICI 182780 (p ≤ 0.05). Similarly, the expression of *Apela* was significantly up-regulated accompanied by 17β-estradiol treatment, whereas recovered significantly to the level of control group in the co-treatment group of 17β-estradiol and

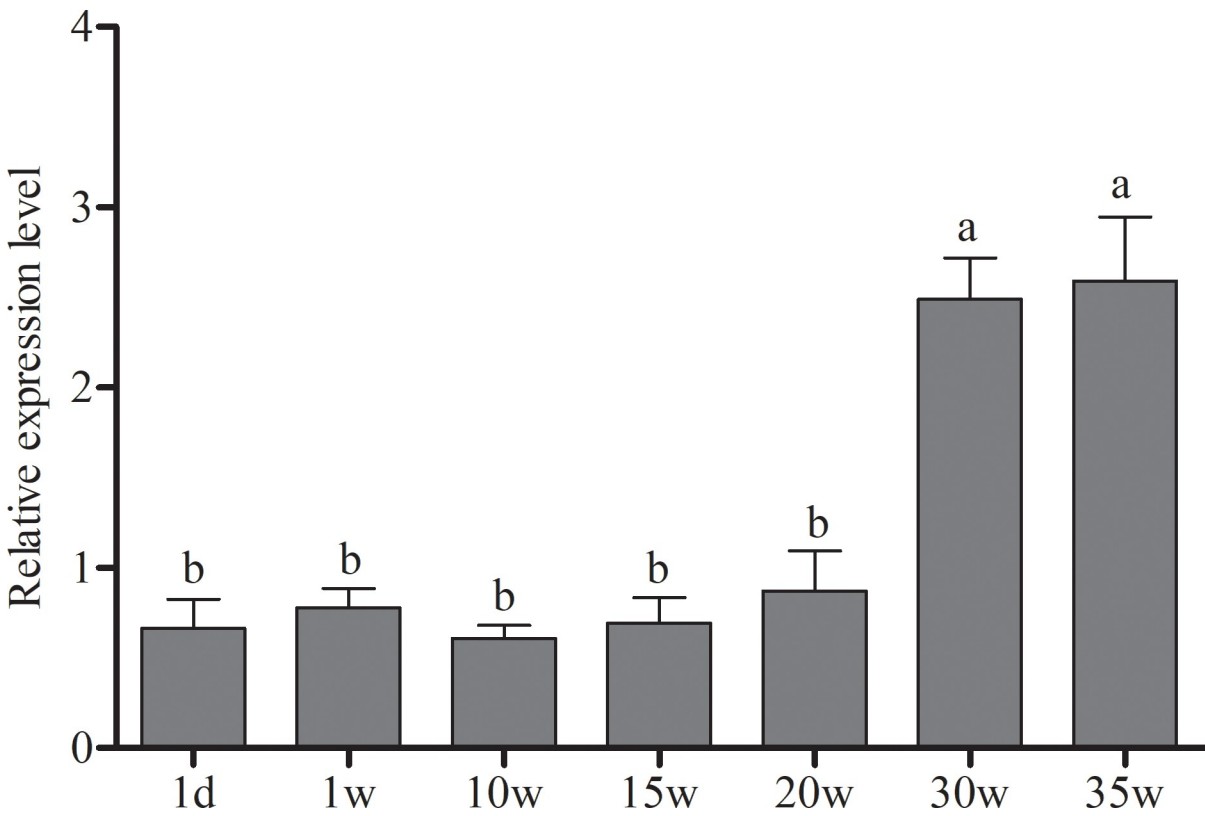

**Fig 4. The relative expression levels of *Apela* gene in the liver of chicken at different growth stages.** 1d means liver tissue of 1day-old chicks. 1w, 10w, 15w, 20w, 30w and 35w represent liver tissue of 1week-, 10week-, 15week-, 20week-, 30week- and 35week-old chicken, respectively. The mRNA levels of *Apela* gene was normalized to the mRNA levels of *β-actin*. Each data point represents the mean ± SEM of 6 chicken. Different lower-case letters mean significant difference (p ≤ 0.05), and the same lower-case letter means no significant difference (p > 0.05).

ER antagonist MPP, tamoxifen or ICI 182780 (p ≤ 0.05). For the estrogen receptor subtype antagonists, ICI 182780 and tamoxifen are antagonists of ERα and ERβ, tamoxifen also induces GPR30, and MPP is a highly selective antagonist of ERα. Therefore, the results implied that the expression of *Apela* in chicken liver was regulated by estrogen through ERα.

## The effect of estrogen on Apela expression in individuals

To further validate the effect of estrogen on chicken *Apela* expression in individuals, 10-week-old hens were treated with different doses of 17β-estradiol for 12 h and 24 h. The serum biochemical index showed estrogen could notably induce the content increase of TG when treating for 12 h (p ≤ 0.05), while the contents of TC and VLDL-c was no significant change (p ≥ 0.05) (Fig 7A). When treating for 24 h, the contents of TG, TC and VLDL-c dramatically increased at 1.0 and 2.0 mg/kg of 17β-estradiol treatment concentrations (p ≤ 0.05) (Fig 7A). Therefore, 17β-estradiol-treated liver tissues for 24 hours were selected for gene expression analysis. The gene expression results showed *ApoVLDLII* was significantly up-regulated expression in a dose-dependent manner accompanied by the increase of 17β-estradiol concentrations (p ≤ 0.05) (Fig 7B). However, when 17β-estradiol treatment concentrations reached 1.0 mg/kg and 2.0 mg/kg levels, *Apela* were induced to be significantly up-regulated in chicken livers (p ≤ 0.05) (Fig 7C).

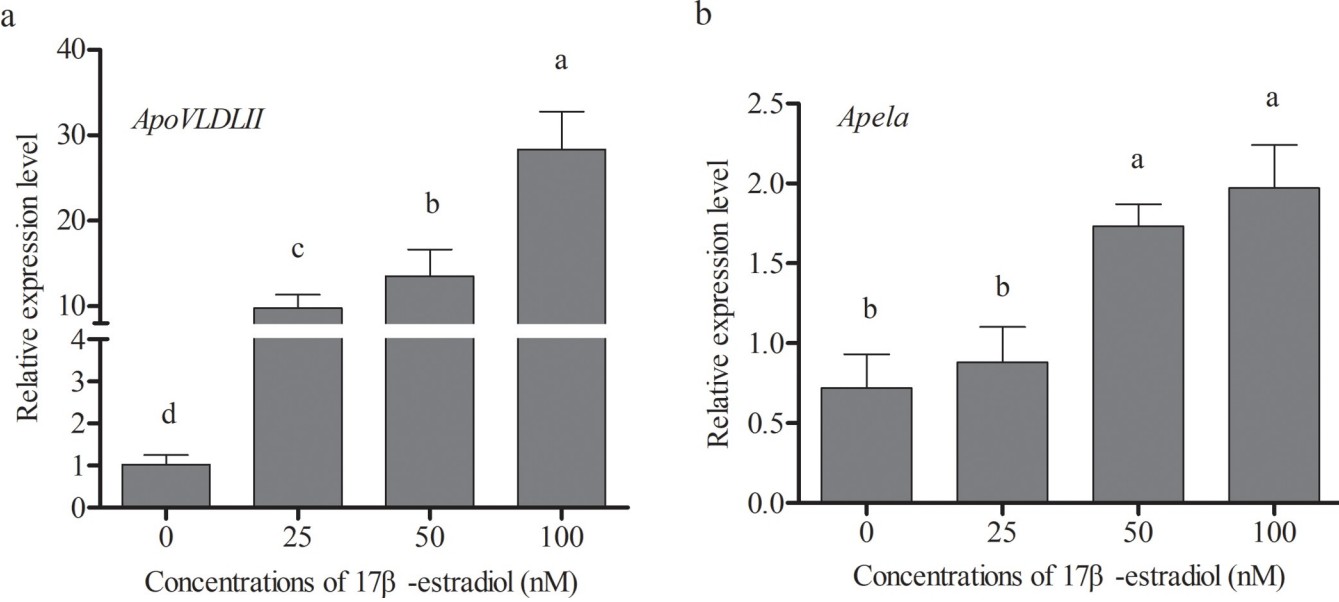

**Fig 5. The effect of 17β-estradiol on the expression of *Apela* gene in chicken embryo hepatocytes.** (a) The mRNA expression of *ApoVLDL II* in hepatocytes. (b) The mRNA expression of *Apela* in hepatocytes. The control groups were treated with solvent ethyl alcohol only, and the treatment groups were treated with 25, 50, 100 nM of 17β-estradiol for 12 h. Each data point represents the mean ± SEM of 6 repeats for each treatment. Different letters mean a significant difference between groups (p ≤ 0.05), and the same letter means no significant difference between groups (p > 0.05).

## Discussion

Previous reports showed that the apelin peptide deficiency in mice didn't cause impact to birth rate or embryonic development [6], while the loss of APJ in mice resulted in a drop of birth rate in the expected Mendelian ratio [7], breaks the hypothesis that apelin was the sole

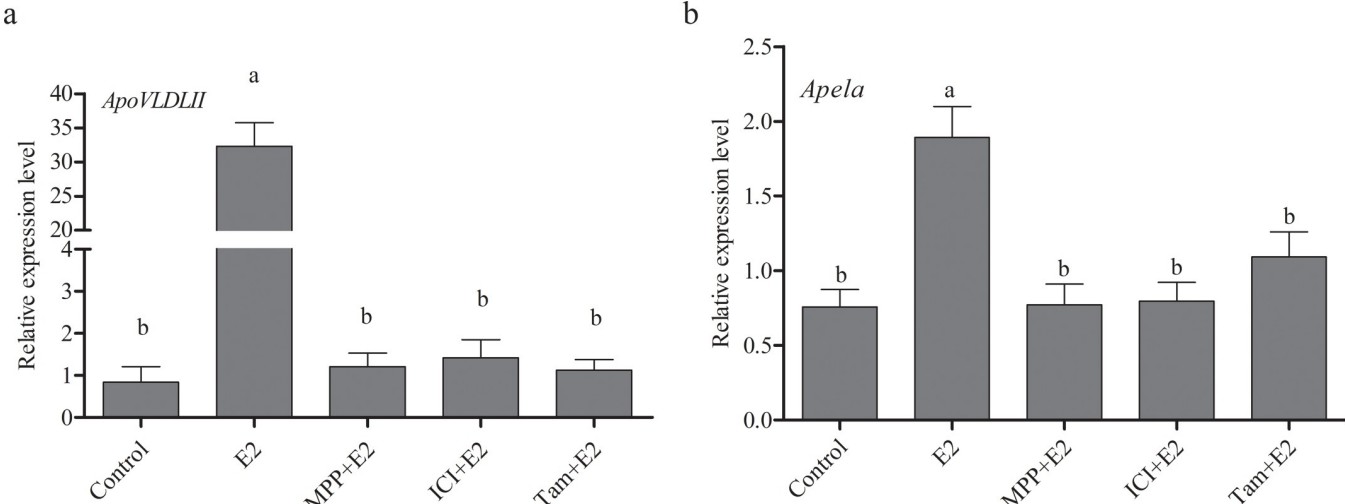

**Fig 6. Effects of different ER subtype antagonists on the expression of the *Apela* gene in chicken embryo hepatocytes.** Effects of different ER subtype antagonists on the expression of the Apela gene in chicken embryo hepatocytes. Control was treated with solvent of the same dose DMSO and supplemental ethyl alcohol. E2 represents the treatment group with 17β-estradiol (100 nM). MPP+E2, ICI+E2 and Tam+E2, represent the co-treatent group with 17β-estradiol and estrogen receptor antagonists MPP (1 μM), ICI (1 μM) and tamoxifen (1 μM), respectively. Each data point represents the mean ± SEM of 6 repeats for each treatment. Different letters mean a significant difference between groups (p ≤ 0.05), and the same letter means no significant difference between groups (p > 0.05).

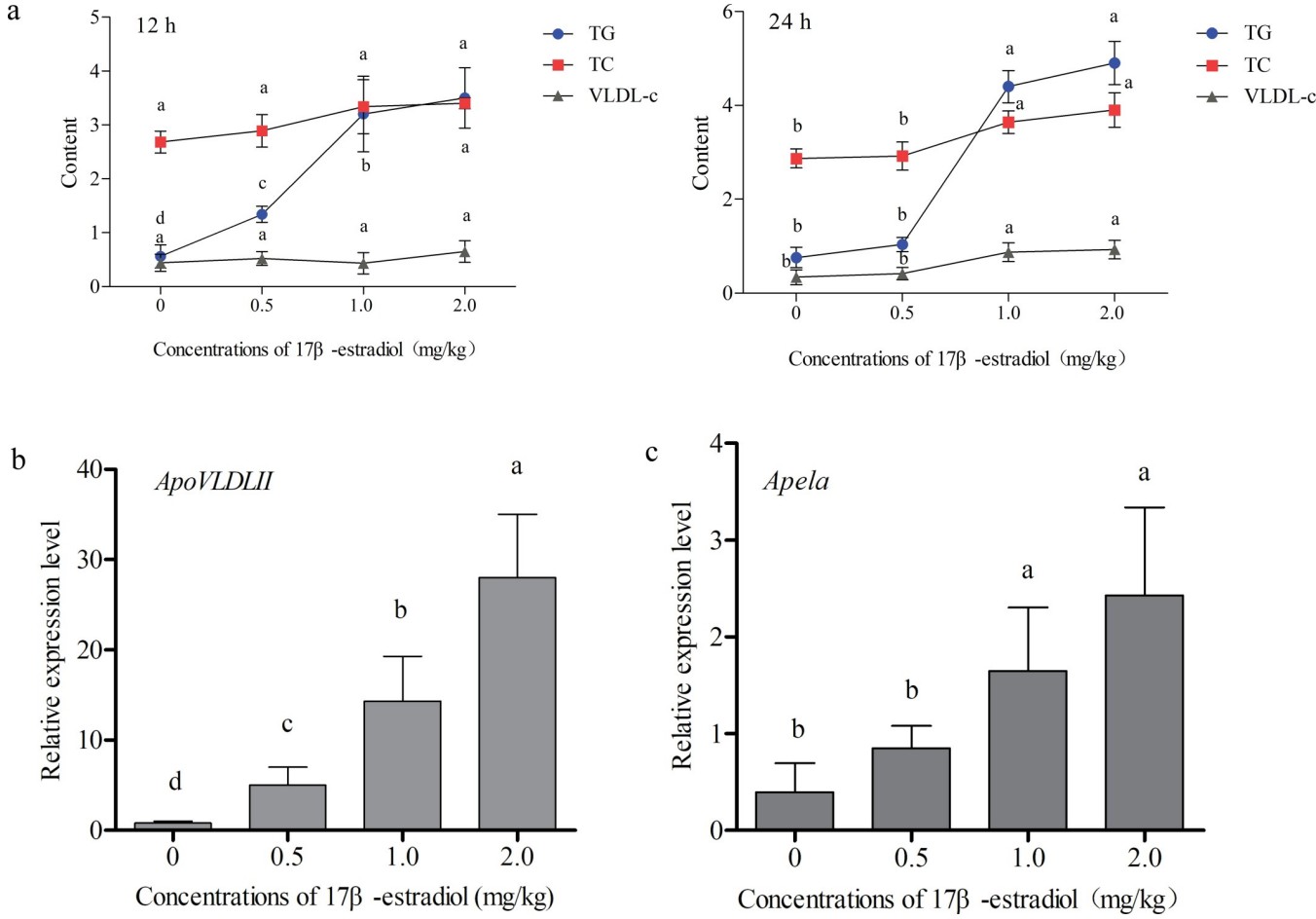

**Fig 7. The effect of 17β-estradiol treatment on the serum biochemical index and liver *Apela* expression in 10-week-old hens.** (a) The contents of TG, TC and VLDL-c in hen serum, accompanied by the change of 17β-estradiol concentrations for 12 and 24 h, respectively. The blue dots represent TG, the red boxes represent TC, and the Grey triangles represent VLDL-c. (b) and (c) The mRNA expression of *ApoVLDLII* and *Apela* in 10-week-old hen liver, accompanied by the change of 17β-estradiol concentrations for 24 h. The control groups (0 mg/kg) were treated with olive oil only, and the treatment groups were treated with 0.5, 1.0 and 2.0 mg/kg of 17β-estradiol. Each data point represents the mean ± SEM of 6 chicken. Different letters mean a significant difference between groups (p ≤ 0.05), and the same letter means no significant difference between groups (p > 0.05).

endogenous ligand of APJ. The discovery and identification of Apela in human embryonic stem cells and the function of its binding APJ confirmed Apela was the second endogenous ligand of APJ [8]. Since then, it brings a blossom on Apela-APJ research. Recently, Apela was detected in the adult heart as a novel endogenous ligand of APJ [33, 36] and was also abundant in human placentas to enhance placental development to prevent preeclampsia [34, 37]. At present, researches on the Apela/APJ pathway and function have mainly focused on human and mouse, no systematic expression and regulation characteristics of *Apela* has been reported in chicken.

Here we cloned the CDs of *Apela* mRNA, analyzed the amino acid sequences and implemented functional domain analysis. The amino acid sequences alignment of Apela among chickens, humans and rats showed that Apela possessed a pair of conserved cysteines, and the last 13 residues were nearly invariant, which was consistent with the analysis of other report [8, 27]. For Apelin, there were various of active protein subunits, such as, Apelin-55, Apelin-17, Apelin-13 and Pyr-Apelin-13 [35, 38]. They all contained the 13 amino acids in the C-precursor protein were thought to be the active units of each subunit [36, 39]. Similar to Apelin,

the Apela amino acids had high similarity in the 13 amino acids [37, 40], so we speculated that the Apela and Apelin were likely to retain similar binding mechanism. However, in this study, the phylogenetic analysis showed that Apelin and Apela were classed into two clusters, which suggested some modest differences of biological functions between these two orthologous genes.

Expression patterns of apela are various in different species, which may associate with its biological functions. In Zebrafish, apela is expressed predominantly in the testes and weakly in other tissues including the intestine, brain and heart [41]. In human beings, expression of apela is associated with pre-implantation human development, and restricted to a few tissues, including cardiovascular tissues and two endocrine organs, the kidneys and placenta in the adult [42]. In rodents, expression of apela can be detected in the stage of zygotic transcription, peaked at the blastocyst stage, and mainly expressed in kidney of adult rat to regulate fluid homeostasis by directly binding APJ to activate the Gi pathway [43]. As for chicken, both our study and other previous study [41] showed that *Apela* is mainly expressed in liver and other two endocrine organs kidney and pancreas. *Apela* mRNA expression exhibited a down-regulation in the peak-laying stage in comparison to that in pre-laying stage in chicken. How does the expression changes of *Apela* in kidney and pancreas tissues correlate to the physiological changes are not clear yet, and need to be clarified by more experiments. However, *Apela* mRNA expression levels in chicken livers at the laying stages are significantly higher than that of pre-laying stages. It was suggested that Apela might play an important role in avian liver lipid metabolism. As we know, the most of *de novo* synthesis of fatty acid occurs in chicken liver [20, 39, 44, 45]. During the laying stages in chicken, proteins and hydrophobic lipids including triacylglycerols, cholesteryl, cholestery esters, phospholipids and free fatty acids are actively synthesized in liver. Further studies focusing on the function of the chicken *Apela* gene in the lipid metabolism of the kidney are needed.

It has been demonstrated that estrogen is a key regulator controlling the lipid synthesis and transport in liver of laying hen [40, 46]. We conducted the estrogen treatment experiments *in vitro* and *in vivo* by using 17β-estradiol. The expression of *Apela* mRNA was increased in a dose-dependent manner both in embryonic primary hepatocytes and liver tissue after 17β-estradiol treatment. This finding indicated that 17β-estradiol could induce the expression of chicken *Apela* gene. It is well known that estrogen stimulates gene expression by binding to its receptors, including nuclear receptors (ERα and ERβ) [41, 42, 47, 48] and a membrane receptor (GPR30) [43, 49]. To further investigate regulatory mechanism of estrogen function on *Apela* expression, the expression levels of *ER α* and *ER β* in the liver of pre- and peak-laying hens were examined first, and found that the expression levels of *ER α* and *ER β* increased with sexual maturation. It should be the true because, in general, the estrogen level of hens rises significantly during the egg-laying period, and estrogen acts on many of genes involved in hepatic metabolisms, e.g. lipid and fatty acid metabolism, by interaction with either ER α or ER β, and certainly other pathways. Then the effects of 17β-estradiol on promoting *Apela* mRNA expression were checked by using the different combinations of 17β-estradiol and ER antagonists, and found that *Apela* expression could be partially inhibited by either the ER α antagonist MPP, or the ER α and ER β antagonists TAM and ICI 182,780. Because MPP is a highly selective antagonist of ERα, and ICI 182, 780 and tamoxifen are antagonist of ERα and ERβ, meanwhile tamoxifen also induces GPR30, a membrane receptor. We speculated that the expression of *Apela* was activated by estrogen in chicken embryo hepatocytes mediated by ERα. Of course, it requires further experimental verification.

During the laying stage, TG and cholesterol are synthesized in large quantities in hen's liver and are assembled by ApoVLDLII and apolipoprotein B into VLDLy, which is a yolk- targeted VLDL, induced by estrogen [41]. The concurrent inductions of avian hepatic lipogenesis and

plasma lipids by estrogen exogenous injection have been examined in immature avian [24]. Our results that the contents of TG, TC, and VLDL-c in serum of 10-week hens stimulated by 17β-estradiol increased significantly, and the liver *ApoVLDLII* expression of the corresponding treatment were also remarkable increase, which is consistent with previous reports. Therefore, the significantly upregulated expression of chicken liver *Apela* induced by 17β-estradiol suggested that *Apela* might involve in hepatic lipid metabolism of laying chicken.

As we usually believed, *Apela* plays its roles via binding APJ. However, previous study demonstrated that *Apela* mRNA has regulatory roles on gene expression depending on species. A previous study reported a non-coding role of *Apela* in promoting p53-mediated DNA damage-induced apoptosis DIA via interaction between heterogeneous nuclear ribonucleoprotein L (hnRNPL) and 3' UTR of *Apela* in embryonic stem cells of mouse origin (mESCs) [19]. However, in human pluripotent embryonic stem cells (hESCs), the mature ELA functions as an endogenous hormonal peptide secreted by hESCs in a paracrine manner and signals through an unknown receptor to the PI3K/AKT pathway to sustain survival and self-renewal of ESCs. We analyzed the the features of 3'UTR sequences of *Apela* between mouse and chicken [42]. The 3'UTRs are of very different lengths and show a low homology between these two orthologous genes, which implied the 3'UTR of *Apela* gene in chicken and mouse may play different roles, but need to be explored fuether.

In conclusion, the CDs of *Apela* mRNA was cloned. Amino acid sequence alignment analysis showed that the last 13 AA residues of Apela were nearly invariant in chicken, human and rat, implicating similarity of Apela binding Apelin receptor between chicken and mammals. Evolutionary analysis showed that *Apela* gene had a distant evolutionary pattern between chickens and mammals, implicating specificity of Apela function in chicken. Furthermore, the *Apela* gene was highly expressed in liver, kidney and pancreas in chicken, and its expression reached a peak in the livers of hens at the peak-laying stages. The expression of the *Apela* gene was induced by estrogen via ERα.

## Supporting information

**S1 Raw image.**
(JPG)

**S1 Fig. The relative expression levels of *Apelin* and *Apelin receptor* gene in the liver of chicken at different growth stages.** 1d means liver tissue of 1day-old chicks. 1w, 10w, 15w, 20w, 30w and 35w represent liver tissue of 1week-, 10week-, 15week-, 20week-, 30week- and 35week-old chicken, respectively. The mRNA levels of *Apelin* and *Apelin receptor* gene was normalized to the mRNA levels of *β-actin*. Each data point represents the mean ± SEM of 6 chicken. Different lower-case letters mean significant difference ($p \leq 0.05$), and the same lower-case letter means no significant difference ($p > 0.05$).
(TIF)

**S2 Fig. The effect of 17β-estradiol on the expression of *Apelin* and *Apelin receptor* gene in chicken embryo hepatocytes.** (a) The mRNA expression of *Apelin* in hepatocytes. (b) The mRNA expression of *Apelin receptor* in hepatocytes. The control groups were treated with solvent ethyl alcohol only, and the treatment groups were treated with 25, 50, 100 nM of 17β-estradiol for 12 h. Each data point represents the mean ± SEM of 6 repeats for each treatment. Different letters mean a significant difference between groups ($p \leq 0.05$), and the same letter means no significant difference between groups ($p > 0.05$).
(TIF)

**S3 Fig. The relative expression levels of *Apelin* and *Apelin receptor* gene in the liver of chicken between 10-week-old juvenile female chicken and 30-week-old sexually mature laying female chicken.** (a, b) The relative expression levels of ER α and ER β, respectively. The mRNA levels of *Apelin* and *Apelin receptor* gene was normalized to the mRNA levels of *β-actin*. Each data point represents the mean ± SEM of 6 chicken. Different lower-case letters mean significant difference ($p \leq 0.05$), and the same lower-case letter means no significant difference ($p > 0.05$).
(TIF)

**S4 Fig. Coding ability analysis and transcriptional factor prediction.** The homology analysis of the 3'UTR sequences of Apela is identical between house mouse and chicken.
(TIF)

## Author Contributions

**Conceptualization:** Wenbo Tan.

**Data curation:** Wenbo Tan.

**Formal analysis:** Wenbo Tan.

**Funding acquisition:** Hang Zheng.

**Investigation:** Hang Zheng.

**Methodology:** Dandan Wang.

**Project administration:** Dandan Wang.

**Resources:** Fangyuan Tian.

**Supervision:** Fangyuan Tian.

**Validation:** Fangyuan Tian, Hong Li.

**Visualization:** Hong Li.

**Writing – original draft:** Wenbo Tan, Hong Li, Xiaojun Liu.

**Writing – review & editing:** Xiaojun Liu.

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
