## [Decision Letter · Decision Letter 0]

5 Jun 2020

PONE-D-20-14270

Expression characteristics and regulatory mechanism of Apela gene in liver of chicken (Gallus gallus)

PLOS ONE

Dear Dr. Liu,

Thank you for submitting your manuscript to PLOS ONE. After careful consideration, we feel that it has merit but does not fully meet PLOS ONE’s publication criteria as it currently stands. Therefore, we invite you to submit a revised version of the manuscript that addresses the points raised during the review process.

We look forward to receiving your revised manuscript.

Kind regards,

Michael Bader

Academic Editor

PLOS ONE

Journal Requirements:

Additional Editor Comments (if provided):

Reviewers' comments:

Reviewer's Responses to Questions

**Comments to the Author**

1. Is the manuscript technically sound, and do the data support the conclusions?

Reviewer #1: Partly

Reviewer #2: Partly

2. Has the statistical analysis been performed appropriately and rigorously? 

Reviewer #1: Yes

Reviewer #2: N/A

3. Have the authors made all data underlying the findings in their manuscript fully available?

Reviewer #1: Yes

Reviewer #2: Yes

4. Is the manuscript presented in an intelligible fashion and written in standard English?

Reviewer #1: Yes

Reviewer #2: Yes

5. Review Comments to the Author

Reviewer #1: In this manuscript Tan et al shew that the expression pattern of apela at different tissues at two time points and found that apela was expressed highly in liver et al. Finally Tan et al wanted to prove the 17β-estradiol induce the expression of apela to involves in liver lipid metabolism. This work is inteseting but far more work is needed to confirm the conclusion (experiments and discussion).

1, why liver et al while not ovary or another related tissues displayed high expression of apela? discussion was needed here.

2, the authors used different ER inhibitor to evaluate which receptor was in the cascade to control apela expression, it looks reasonable. While we suggest the author to check the expression of the different receptors in different tissues at the two time points, that is benefit to get the conclusion.

3, the authors wanted to show apela mediates ER to regulate liver lipid metablism, and used the word "likely", but I suggest to do more work to confirm it since it is easy to do. If not we suggest not to discuss this point in the manuscript. You know there is a big gap here.

4, the disscussion only list the experiment again while not discuss the question about this study. In addition, we suggested the authors discuss the study within a deep way.

Reviewer #2: The authors reported the cloning of Apela/Elabela gene in chicken, its expression across the tissues and inducible expression in the liver of hens or hepatocytes at the peak-laying stage or by 17beta-estradiol treatment. They concluded that Apela may be involved in liver lipid metabolism of laying chicken through estrogen-induced regulation. The cloning of Apela in chicken was reported 2 years ago, in which Apelin and Apelin receptor were also well characterized (PMID: 30631305). What’s new here is that inducible expression of Apela in the liver and its correlation with ApoVLDLII expression.

Comments

1. Apelin (Apln) and Apelin receptor (Aplnr) should be examined in the liver of hens and hepatocytes at the peak-laying stage and by 17beta-estradiol treatment, respectively.

2. A key question is whether Apela gene in chicken express the peptide. Since the 3’UTR of Apela mRNA has regulatory roles on gene expression depending on species (PMID: 25936916), it would be possible that chick Apela may only serve as a non-coding RNA.

3. It should be clarified whether Apela peptide indeed regulate lipid metabolism in the liver of laying chicken. It is also possible that Apela may regulate lipid metabolism in other organs in endocrine manner. These points should be investigated experimentally.

4. Compared with induction of ApoVLDLII, increase of Apela expression is modest at the peak-laying stage. Down-regulation of Apela expression in pancreas is more robust. This is also needed to examine more carefully together with expression of Apln and Aplnr.

6. PLOS authors have the option to publish the peer review history of their article (what does this mean?). If published, this will include your full peer review and any attached files.

Reviewer #1: No

Reviewer #2: No

---

## [Author Response · Author response to Decision Letter 0]

26 Jul 2020

Dear Dr. Bader,

We thank both you and reviewers for your suggestions and comments on our manuscript entitled “Expression characteristics and regulatory mechanism of Apela gene in liver of chicken (Gallus gallus)” (ID: PONE-D-20-14270). We have amended our manuscript accordingly and marked the revised portion in red. The point by point responses to the comments and suggestions raised by reviewers are listed below. We hope that the revised manuscript is acceptable for publication in your Journal PLOS ONE.

Should you have any further questions, please don’t hesitate to contact us.

Yours sincerely,

Xiaojun Liu & Wenbo Tan, on the behalf of all the co-authors

College of Animal Science and Veterinary Medicine 

Henan Agricultural University

Zhengzhou 450046

China

Tel: 0086-371-63555618

Fax: 0086-371-63558180

Email: xjliu2008@hotmail.com

603679737@qq.com

The point by point responses to the comments raised by reviewers

Reviewer #1: In this manuscript Tan et al shew that the expression pattern of apela at different tissues at two time points and found that apela was expressed highly in liver et al. Finally Tan et al wanted to prove the 17β-estradiol induce the expression of apela to involves in liver lipid metabolism. This work is inteseting but far more work is needed to confirm the conclusion (experiments and discussion).

Author response: Thank you very much for your comments on the work. To the best of our knowledge, this is the first time to report the expression characteristics and regulatory mechanism of Apela gene in liver of chicken. We have carefully amended our manuscript according to your comments and suggestions, and hope the revised version is satisfied. 

1, why liver et al while not ovary or another related tissues displayed high expression of apela? discussion was needed here. 

Author response: Thank you very much for your suggestion. According to the studies on apela expression so far, expression patterns of apela are various in different species, which may associate with its biological functions. In Zebrafish, apela is expressed predominantly in the testes and weakly in other tissues including the intestine, brain and heart. In human beings, expression of apela is associated with pre-implantation human development, and restricted to a few tissues, including cardiovascular tissues and two endocrine organs, the kidneys and placenta in the adult. In rodents, expression of apela can be detected in the stage of zygotic transcription, and peaked at the blastocyst stage. As for chicken, both our study and other previous study showed that apela is mainly expressed in liver and other two endocrine organs kidney and pancreas. We have amended our manuscript by adding the above information with corresponding references. Please see details in the revised manuscripts in line 375-381, line 382-388 and line 393-394.

2, the authors used different ER inhibitor to evaluate which receptor was in the cascade to control apela expression, it looks reasonable. While we suggest the author to check the expression of the different receptors in different tissues at the two time points, that is benefit to get the conclusion.

Author response: Thank you very much for your valuable suggestions. Accordingly, we checked the expression levels of ER α and ER β in the liver of pre-laying hens (20 weeks old) and peak-laying hens (30 weeks old), and found that the expression levels of ER α and ER β increased with sexual maturation. It should be the true because, in general, the estrogen level of hens rises significantly during the egg-laying period, and estrogen acts on many of genes involved in hepatic metabolisms, e.g. lipid and fatty acid metabolism, by interaction with either ER α or ER β, and certainly other pathways. We have added the results in line 303-306 and 402-411 of text and Supplement Fig. 3 in the revised manuscript.

3, the authors wanted to show apela mediates ER to regulate liver lipid metablism, and used the word "likely", but I suggest to do more work to confirm it since it is easy to do. If not we suggest not to discuss this point in the manuscript. You know there is a big gap here.

Author response: Thank you very much for your valuable suggestions. To make the description more precise, we have removed all the parts related the "likely" in the text of the revised manuscript.

4, the discussion only list the experiment again while not discuss the question about this study. In addition, we suggested the authors discuss the study within a deep way.

Author response: Thank you very much for your comments. We have carefully amended the discussion section of our manuscript by adding more information about the evolution and function of the gene in details on line 371-374, 375-381, 382-388, and 393-394. We also added the discussion about the mechanism of estrogen regulating apela expression in details in line 402-411. In addition, we discussed the coding ability of Apela in details as well in line 426-437.

Reviewer #2: The authors reported the cloning of Apela/Elabela gene in chicken, its expression across the tissues and inducible expression in the liver of hens or hepatocytes at the peak-laying stage or by 17beta-estradiol treatment. They concluded that Apela may be involved in liver lipid metabolism of laying chicken through estrogen-induced regulation. The cloning of Apela in chicken was reported 2 years ago, in which Apelin and Apelin receptor were also well characterized (PMID: 30631305). What’s new here is that inducible expression of Apela in the liver and its correlation with ApoVLDLII expression.

Author response: Thank you very much for your comments on our work. Most of the work presented in the manuscript was being done before 2017 by a master student, who obtained her degree on July 2017. During the preparation of the manuscript, many related research papers published. Some of our work lost their novelty. Therefore, we focused only on “expression characteristics and regulatory mechanism of Apela gene in liver of chicken” in the manuscript. The related results are added in line 109.

Comments

1. Apelin (Apln) and Apelin receptor (Aplnr) should be examined in the liver of hens and hepatocytes at the peak-laying stage and by 17beta-estradiol treatment, respectively.

Author response: Thank you very much for your suggestion. We have examined the expression of Apelin (Apln) and Apelin receptor in the liver of hens at the peak-laying stage and hepatocytes treated by 17beta-estradiol, respectively. The related results are added in line 256, 266-268, 292-293 and Supplementary Fig. 1 and 2.

2. A key question is whether Apela gene in chicken express the peptide. Since the 3’UTR of Apela mRNA has regulatory roles on gene expression depending on species (PMID: 25936916), it would be possible that chick Apela may only serve as a non-coding RNA.

Author response: Thank you very much for your valuable comments. We refer to the results of this paper (PMID: 25936916) and analyzed 3’UTR of chicken Apela mRNA. We found that Apela mRNA has regulatory roles on gene expression depending on species.The 3’UTRs are of very different lengths and show a low homology between these two orthologous genes, which implied the 3'UTR of Apela gene in chicken and mouse may play different roles. Of course, it requires further experimental verification. The result is added in line 92-95, 426-437 and Supplement Fig. 4 in the revised manuscript. 

3. It should be clarified whether Apela peptide indeed regulate lipid metabolism in the liver of laying chicken. It is also possible that Apela may regulate lipid metabolism in other organs in endocrine manner. These points should be investigated experimentally.

Author response: Thanks for your comments and suggestions. Indeed, we don’t have direct solid evidences to confirm that Apela peptide regulates lipid metabolism in the liver of laying chicken or other organs in endocrine manners. Since the manuscript focuses on “expression characteristics and regulatory mechanism of Apela gene in liver of chicken”, we have removed the related statements in the revised manuscript as suggested by other reviewer.

4. Compared with induction of ApoVLDLII, increase of Apela expression is modest at the peak-laying stage. Down-regulation of Apela expression in pancreas is more robust. This is also needed to examine more carefully together with expression of Apln and Aplnr.

Author response: Thank you very much for your valuable comments. The expression of Apela in pancreas did appear significantly changes, we will focus on the role of apela in pancreas in future research, and carry out in-depth study on its biological function. The relevant description is added in line 402-411.

---

## [Decision Letter · Decision Letter 1]

25 Aug 2020

Expression characteristics and regulatory mechanism of Apela gene in liver of chicken (Gallus gallus)

PONE-D-20-14270R1

Dear Dr. Liu,

We’re pleased to inform you that your manuscript has been judged scientifically suitable for publication and will be formally accepted for publication once it meets all outstanding technical requirements. One reviewer did not like your answer to the question on the possible non-coding function of the APELA-mRNA you describe. But you mention the publication (ref 19) about noncoding functions of the mRNA in human ESCs and the paper (ref 42) showing that it also produces a peptide in the same cells, which is sufficient to address this issue in my opinion.

Kind regards,

Michael Bader

Academic Editor

PLOS ONE

Additional Editor Comments (optional):

Reviewers' comments:

Reviewer's Responses to Questions

**Comments to the Author**

1. If the authors have adequately addressed your comments raised in a previous round of review and you feel that this manuscript is now acceptable for publication, you may indicate that here to bypass the “Comments to the Author” section, enter your conflict of interest statement in the “Confidential to Editor” section, and submit your "Accept" recommendation.

Reviewer #1: All comments have been addressed

Reviewer #2: (No Response)

2. Is the manuscript technically sound, and do the data support the conclusions?

Reviewer #1: Partly

Reviewer #2: No

3. Has the statistical analysis been performed appropriately and rigorously? 

Reviewer #1: Yes

Reviewer #2: No

4. Have the authors made all data underlying the findings in their manuscript fully available?

Reviewer #1: Yes

Reviewer #2: Yes

5. Is the manuscript presented in an intelligible fashion and written in standard English?

Reviewer #1: Yes

Reviewer #2: Yes

6. Review Comments to the Author

Reviewer #1: Your group found that the interesting expression pattern of apela/aplnr in chicken, and checked tdhe relationship between them and ER signaling, it is intersting and useful for the futher study. while the function of apela in liver metabolism need far more work to address.

Reviewer #2: Apela is a gene encoding a peptide ligand for Apelin receptor, but the authors did not clarify whether their identified gene is Apela or related pseudogene or non-coding RNA. The authors did not address the comments raised by this reviewer experimentally.

7. PLOS authors have the option to publish the peer review history of their article (what does this mean?). If published, this will include your full peer review and any attached files.

Reviewer #1: **Yes: **Sizhou Huang

Reviewer #2: No

---

## [Editor Report · Acceptance letter]

31 Aug 2020

PONE-D-20-14270R1 

Expression characteristics and regulatory mechanism of Apela gene in liver of chicken (Gallus gallus) 

Dear Dr. Liu:

I'm pleased to inform you that your manuscript has been deemed suitable for publication in PLOS ONE. Congratulations! Your manuscript is now with our production department. 

Kind regards, 

on behalf of

Prof. Michael Bader 

Academic Editor

PLOS ONE